# Abundance and Updated Distribution of *Aedes aegypti* (Diptera: Culicidae) in Cabo Verde Archipelago: A Neglected Threat to Public Health

**DOI:** 10.3390/ijerph17041291

**Published:** 2020-02-17

**Authors:** Silvânia Da Veiga Leal, Isaias Baptista Fernandes Varela, Aderitow Augusto Lopes Gonçalves, Davidson Daniel Sousa Monteiro, Celivianne Marisia Ramos de Sousa, Maria da Luz Lima Mendonça, Adilson José De Pina, Maria João Alves, Hugo Costa Osório

**Affiliations:** 1Laboratório de Entomologia Médica, Instituto Nacional de Saúde Pública, Largo do Desastre da Assistência, Chã de Areia, Praia 719, Cabo Verde; IsaiasB.Varela@insp.gov.cv (I.B.F.V.); Aderitow.Goncalves@insp.gov.cv (A.A.L.G.); Davidson.Monteiro@insp.gov.cv (D.D.S.M.); Celivianne.Sousa@insp.gov.cv (C.M.R.d.S.); Mariadaluz.Lima@insp.gov.cv (M.d.L.L.M.); 2Programa de Pré-Eliminação do Paludismo, CCS-SIDA, Ministério da Saúde e da Segurança Social, Varzea, Praia 855, Cabo Verde; Adilson.Pina@ccssida.gov.cv; 3Instituto Nacional de Saúde Doutor Ricardo Jorge, Centro de Estudos de Vectores e Doenças Infecciosas, Avenida da Liberdade 5, 2965-575 Águas de Moura, Portugal; m.joao.alves@insa.min-saude.pt (M.J.A.); hugo.osorio@insa.min-saude.pt (H.C.O.); 4Instituto de Saúde Ambiental, Faculdade de Medicina da Universidade de Lisboa, Av. Prof. Egas Moniz, Ed. Egas Moniz, Piso 0, Ala C, 1649-028 Lisboa, Portugal

**Keywords:** *Aedes aegypti*, arboviruses, larval index, surveillance, control, Cabo Verde

## Abstract

*Background:* Mosquito-borne viruses, such as Zika, dengue, yellow fever, and chikungunya, are important causes of human diseases nearly worldwide. The greatest health risk for arboviral disease outbreaks is the presence of the most competent and highly invasive domestic mosquito, *Aedes aegypti*. In Cabo Verde, two recent arbovirus outbreaks were reported, a dengue outbreak in 2009, followed by a Zika outbreak in 2015. This study is the first entomological survey for *Ae. aegypti* that includes all islands of Cabo Verde archipelago, in which we aim to evaluate the actual risk of vector-borne arboviruses as a continuous update of the geographical distribution of this species. *Methods:* In order to assess its current distribution and abundance, we undertook a mosquito larval survey in the nine inhabited islands of Cabo Verde from November 2018 to May 2019. Entomological larval survey indices were calculated, and the abundance analyzed. We collected and identified 4045 *Ae. aegypti* mosquitoes from 264 positive breeding sites in 22 municipalities and confirmed the presence of *Ae. aegypti* in every inhabited island. **Results**: Water drums were found to be the most prevalent containers (*n* = 3843; 62.9%), but puddles (*n* = 27; 0.4%) were the most productive habitats found. The overall average of the House, Container, and Breteau larval indices were 8.4%, 4.4%, and 10.9, respectively. However, 15 out of the 22 municipalities showed that the Breteau Index was above the epidemic risk threshold. *Conclusion:* These results suggest that if no vector control measures are considered to be in place, the risk of new arboviral outbreaks in Cabo Verde is high. The vector control strategy adopted must include measures of public health directed to domestic water storage and management.

## 1. Introduction

Mosquito-borne arboviral diseases are of global importance. Zika, dengue fever, and chikungunya are currently the most challenging arboviruses to international public health, despite control program efforts and research in new control methodologies [1,2,3]. Four billion people live in geographic areas suitable for dengue virus transmission alone [4,5,6]. The presence and abundance of vector mosquitoes associated with these diseases are the key points for the health risks of arboviral disease outbreaks. Globally, *Aedes aegypti* (Linnaeus, 1762) (=*Stegomyia aegypti*) is the primary vector of all these viruses, followed by other *Aedes* species, namely *Aedes albopictus* (Skuse, 1895) (=*Stegomyia albopicta*), which are a competent and epidemiologically significant species [1]. *Ae. aegypti* is the invasive mosquito species that have caused the most human casualties worldwide. They are a highly anthropophilic, peridomestic, day biting species that usually breed in artificial sites inside or around dwellings [7,8,9,10].

In Cabo Verde, two mosquito-borne virus outbreaks were recently reported for the first time: a dengue outbreak in 2009, with more than 21,000 notified cases, including 174 cases of dengue hemorrhagic fever and four reported deaths. This was followed by a Zika outbreak in 2015, with more than 7500 notified cases and 18 associated cases of microcephaly [11,12,13]. This was the first time that a Zika strain associated with these neurological damages in infants was detected in Africa [12]. In Cabo Verde, *Ae. aegypti* is so far the only mosquito vector associate of these arboviruses.

The archipelago of Cabo Verde is located on the west coast of Africa, and is composed of 10 islands clustered in two groups: the Barlavento group (comprising the islands of Santo Antão, São Vicente, Santa Luzia, São Nicolau, Sal, and Boavista) and the Sotavento group (comprising the islands of Maio, Santiago, Fogo, and Brava). However, each island has specific topography and displays differences in microclimate and vegetation. Historically, the topography and geographical location of Cabo Verde has promoted and allowed for the active movement of population and goods [14]. This increases the risk of pathogen circulation through infected travelers, which can cause the emergence or re-emergence of arboviral diseases if competent mosquito vectors are present and vector capacity is high [15]. In 2015, Cabo Verde had a passenger volume of more than 7000 travelers from Zika-affected countries, including direct flights from Brazil [13]. The modification of the environment by anthropic actions, disordered urban planning, population growth, and emergent factors related to the globalization process and climate change affect the bionomics of mosquito vectors, increasing their vectorial capacity [16,17,18].

In the archipelago of Cabo Verde, 11 mosquito species belonging to five genera were reported: *Aedes caspius* (Pallas, 1771)*, Ae. aegypti* (Linnaeus, 1762), *Anopheles pretoriensis* (Theobald, 1903), *Anopheles arabiensis* (Patton, 1905), *Culex bitaeniorhynchus* (Giles, 1901), *Culex quinquefasciatus* (Say, 1823), *Culex pipiens* (Linnaeus, 1758)*, Culex perexiguus* (Theobald, 1903), *Culex tritaeniorhynchus* (Giles, 1901), *Lutzia tigripes* (de Grandpre and de Charmoy, 1901)*,* and *Culiseta longiareolata* (Macquart, 1838) [19,20,21,22]. Entomological surveys in Cabo Verde started in the 1920s and *Ae. aegypti* was reported for the first time by Sant’Anna in 1931 on São Vicente island [19]. The data from these pioneer surveillance operations were compiled with the last countrywide mosquito survey [19]. In 2007, another entomological survey was carried out on the four islands of the Sotavento group: Maio, Santiago, Fogo, and Brava. *Ae. aegypti* was detected on Santiago, Fogo, and Brava [21]. In 2011, an entomological survey was carried out in Santiago, where *Ae. aegypti* was reported in the municipalities of Praia and Tarrafal [22]. Phylogeographic and population genetic studies of Cabo Verde’s *Ae. aegypti* population suggested an ancient West African origin, most likely from Senegal, and a population belonging to the subspecies *formosus* [23]. Integrated vector control measures, including strategies of source reduction by breeding site elimination, biological control with fish, and chemical control with insecticides, have been used toward controlling malaria and dengue, *Anopheles arabiensis,* and *Ae. aegypti*, respectively [24]. Regarding insecticide susceptibility, knockdown resistance (kdr) mutations, genetic mutation conferring resistance to dichlorodiphenyltrichloroethane (DDT), and pyrethroids insecticides were not found in 2007 and 2010 *Ae. aegypti* samples. This is in line with insecticide susceptibility tests performed on *Ae. aegypti* from Santiago Island during the dengue outbreak in 2009 [23,25]. However, the situation changed in 2012 and 2014, with the first reports of resistance to these insecticides [26].

One of the tools used in *Ae. aegypti* surveillance is the determination of *Stegomyia* indices, namely the House Index (HI), Container Index (CI), and Breteau Index (BI). These indices measure the abundance, spatial distribution, and provide information about areas or periods of mosquito population growth [27,28,29,30,31].

Most of the studies on *Ae. aegypti* based on *Stegomyia* indices support a significant association between these and the transmission risk of arboviruses [32,33,34,35,36]. In studies that did not observe this association, mosquito and human migration were considered as possible factors that affected the lack of association [37,38,39]. Hence, knowledge of these indices allowed for the timely application of control measures and strategies [40].

In this context, we aimed to evaluate the actual risk of vector-borne arboviruses in Cabo Verde based on the *Stegomyia* indices, as a continuous update of the geographic distribution of *Ae. aegypti*. To our knowledge, this study represents the first entomological survey for this species that includes all islands of Cabo Verde archipelago.

## 2. Methods

### 2.1. Study Area

We collected mosquito larvae in the 22 municipalities of Cabo Verde, a volcanic archipelago with an area of 4033 km^2^ located about 550 km off the coast of Senegal. The archipelago consists of 10 islands, nine of which are inhabited with approximately 537,660 inhabitants (Figure 1). It has an arid and semi-arid climate, warm and dry, with an average annual temperature of around 25 °C, and low rainfall. Two seasons can be identified: the dry season, from December to June, and the rainy season, from August to October [41,42].

In 2010, 141,762 accommodations, including 114,469 buildings, were registered in the country. Of those, 94,894 (82.9%) have one division/room, 10,646 (9.3%) have two divisions/rooms, and 6983 (6.1%) are buildings with three or more rooms. Of the total buildings, 74,404 (65%) are finished, while the remaining are under construction. Regarding the type of habitat, 44,185 (38.6%) of the houses are in urban areas and 30,449 (26.6%) in rural areas [43].

More than 95% of the population use conventional material for construction of their houses, 3.9% use non-conventional material, and 1.3% use a thatched roof, brass, drum plates, or others for cover [44,45].

The most common pavement types (99.4%) are cement and mosaic, and only 0.6% are clay or other [46]. In terms of wall cladding, 66.7% are plastered and painted, while just over 16% do not have any type of coating. Regarding the ceilings, most (79.3%) use reinforced concrete terraces [42].

One-third of Cape Verdeans do not have access to public water [45] and for those who do have access, the distribution is irregular, leading to water storage inside and outside of the homes.

In rural and semi-rural areas, pig pens or henneries are found around the houses, from which additional income is obtained [47].

### 2.2. Entomological Collections and Sampling Methodology

From November 2018 to May 2019, mosquito larvae were collected in all municipalities of Cabo Verde. We selected the sampling area with each municipal health delegation team, according to the high incidence history of mosquito-borne diseases, *Ae. aegypti* densities, and the human population. The houses were selected randomly, both in rural and urban areas. In urban areas with two parallel rows of houses, the selection was made by choosing a first house and then skipping four houses, counting a zigzag pattern. All containers, or potential breeding sites with water for larvae, were inspected and recorded (container type, position, vegetation, and sun exposure). The collected larvae were transported to the National Institute of Public Health (INSP) Medical Entomology Laboratory for morphological identification.

### 2.3. Morphological Identification

Larvae and reared adult mosquitoes were morphologically identified as *Ae. aegypti* under a stereomicroscope, according to the identification keys of Ribeiro et al. [19,48,49,50]. The larvae were mounted on slides with 2% glycerinated Hoyer’s medium, and adult reared mosquitoes, and stored at −20 °C for further molecular and genetic analysis.

### 2.4. Statistical Analysis

We compiled the data into a Microsoft Excel database. For the data analysis, the continuous variables were expressed in measures of central tendency and dispersion, and the categorical ones in simple frequency. The chi-square test was used to determine the association between the presence of *Ae. aegypti* and the type of breeder/container (type, position, and physical characteristics). We considered positive breeding sites for *Ae. aegypti* where there was at least one larva. The level of significance for statistical analysis was 0.05. We used IBM SPSS Statistics 20 (International Business Machines Corporation, New York City, NY, USA) to analyze the data.

Larval indices were calculated, namely, HI, CI, and BI. The maps were drawn using ArcGIS 10.6 (Environmental Systems Research Institute, Redlands, CA, USA).

## 3. Results

A total of 2612 houses were surveyed in the 22 municipalities of Cabo Verde, and 6113 containers were inspected. Of these, immature mosquitoes were detected in 7.5% (*n* = 458) of all inspected containers, the majority (84.3%; *n* = 386) located outdoors, and the others (15.7%; *n* = 72) indoors. No significant differences were observed in the distribution of indoor/outdoor and positive/negative breeding sites (*p* > 0.05 in both cases) (Table 1).

Of the 458 containers with immature mosquitoes, 57.6% (*n* = 264) were positive for *Ae. aegypti*, of which 20.8% (*n* = 55) were found inside dwellings, and 79.2% (*n* = 209) were found outside.

A total of 4045 *Ae. aegypti* larvae and pupae were collected from the 264 *Aedes*-positive containers found in all 22 of the country’s municipalities (Table 2). Other mosquito species found across the survey were *Aedes caspius*, *Anopheles pretoriensis*, *Anopheles arabiensis*, *Culex bitaeniorhynchus*, *Culex pipiens* s.l. (*Culex quinquefasciatus*, *Culex pipiens*), *Culex tritaeniorhynchus*, *Lutzia tigripes*, and *Culiseta longiareolata*. *Culex pipiens* s.l. and *An. pretoriensis* were found sharing the same breeding sites with *Ae. aegypti*.

Water drums (50–200 L) were the most common breeding sites (62.9%; *n* = 3843), followed by tanks (1000–5000 L) (12.4%; *n* = 756), cisterns (5000–10,000 L) (6.3%; *n* = 387), and other types of containers that comprised 18.4% of total breeding sites. From all inspected containers, 458 (7.5%) were positive, and the productivity proportion analysis for each container showed puddles as the most productive habitat (33%; *n* = 9), followed by flowerpots (21%; *n* = 47), other (18%; *n* = 53), and tanks (12%; *n* = 91). All other types of containers showed that productivity equaled less than 10% (Figure 2).

The four most productive containers for *Ae. aegypti* were water drums (61.4%), flowerpots (14.4%), tanks (8.0%), and buckets (3.8%) (Table 2). A positive association between the container type and the presence of *Ae. aegypti* was observed (χ^2^ = 133.816, *p* < 0.001). The frequency of other breeding sites was relatively low, 3% for cisterns and other containers, and lower for drinking fountains, puddles, pots, and tires (Table 2).

### Entomological Indices

The average HI, CI, and BI were 8.4%, 4.4%, and 10.9, respectively. Fifteen out of the 22 studied municipalities presented a BI above five. The maximum values were in the municipality of Brava (HI = 25%; CI = 9.7% and BI = 30.2) and the minimum values, below 1%, were found in Sal. In the municipality of Santa Catarina (Fogo island), despite no *Ae. aegypti* larvae being found, adults were recorded during the survey (Table 3).

## 4. Discussion

This study represents the first archipelago-wide analysis of the *Ae. aegypti* breeding sites in Cabo Verde, which are exclusively domestic containers. This domestic mosquito is one of the most important arthropod vectors of arboviruses worldwide, namely dengue, Zika, chikungunya, and yellow fever [51]. Although there has been an ongoing focus on vaccine development for prevention of these diseases, vector control has been the key strategy to control or prevent the transmission of mosquito-borne arbovirus infections [52]. In previous studies, *Ae. aegypti* populations infected with DENV-2 and DENV-4 were found in Cabo Verde, with high vector competence to transmit DENV-2 and DENV-3, and to be infected with and transmit chikungunya and yellow fever [53,54,55].

In Cabo Verde, recent entomological data on the mosquito species distribution in the nine inhabited islands are missing, and data regarding *Ae. aegypti* are scarce. Although several factors influence a breeding site’s availability and mosquito distribution, in this study, *Ae. aegypti* was mostly found in water drums used in water storage by the population, which corroborates previous results [56,57,58,59,60].

High vector density and susceptible human population are key factors to arbovirus disease outbreaks. Between these two, the first one is the major contributor and can be estimated by *Aedes* indices, such as CI, HI, and BI [61]. These larval indices provide useful information to plan, evaluate, and monitor the efficacy of vector control interventions. The BI is the most used, considering the number of positive containers and searched houses. We noticed variation in the indices among the municipalities, with some municipalities showing BI as high as 31.9 and others 0. This variation is highly dependent on a container’s availability, which can be affected by numerous factors, such as seasonality (rainy or dry season), local population habits, customs, and traditions, and local microclimate. The irregularities in the water distribution force people to store water in containers; thus, this factor plays an important role in the ecology of larval mosquito habitats [59]. Positive containers found outside of dwellings were associated with domestic animals and agriculture (not statistically tested), but further studies should be done to approve or disprove this claim.

In this study, the fieldwork occurred during the dry season and we observed that *Ae. aegypti* is extremely adapted to domestic habitats. It is also important to note that no correlation was found between indoor and outdoor containers in this study.

Thirteen municipalities presented a BI above the epidemic risk threshold [40,62]. Our results suggest that *Ae. aegypti* is well established in all archipelago islands, and several municipalities in Cabo Verde are at risk of arboviral disease outbreaks.

## 5. Conclusions

*Aedes aegypti* is a major threat to public health in Cabo Verde, considering the values of larval indices found in this study associated with previous studies showing *Ae. aegypti* vector competence to diseases registered and not registered in Cabo Verde [54,63], as well as resistance to insecticides in the archipelago [25,26]. Nevertheless, more studies are crucial to evaluate this species’ resistance to more insecticides used in the public health context. We also recommend implementation of a countrywide vector control strategy with environmental management and modification, according to general international guidelines [31]. A program for monitoring *Ae. aegypti,* carried out by each municipality’s health delegation, with the support of the Ministry of Health, is also desirable.

## Figures and Tables

**Figure 1 ijerph-17-01291-f001:**
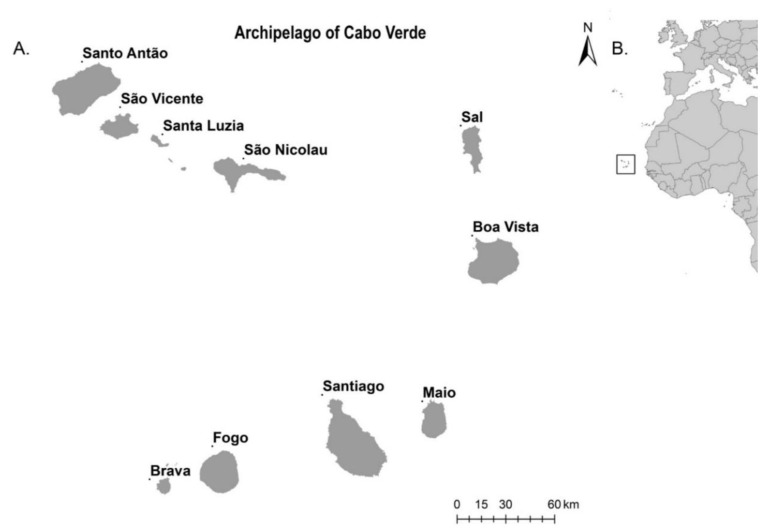
Geographic localization of Cabo Verde archipelago (**A**), islands’ distribution in the archipelago; (**B**), regarding the West Coast of Africa).

**Figure 2 ijerph-17-01291-f002:**
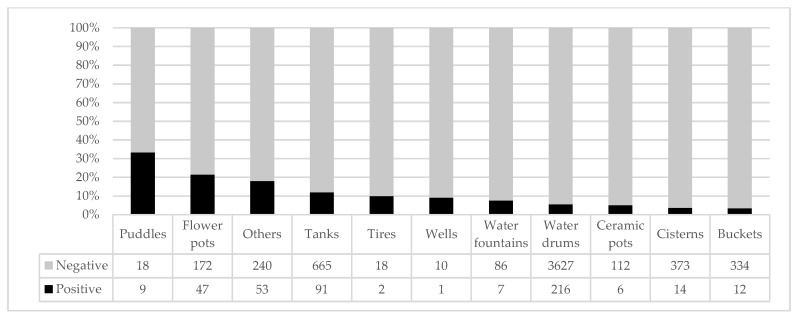
Number and proportion of positive and negative breeding sites inspected during the study.

**Table 1 ijerph-17-01291-t001:** Number of houses, inspected containers, its position (indoor/outdoor), and total number of containers with mosquitoes’ larvae.

Island	Municipalities	All Species			*Ae. aegypti*			Total of Inspected Houses	Total of Inspected Containers
		Positive Houses	Positive Containers		Positive Houses	Positive Containers			
			Indoor	Outdoor		Indoor	Outdoor		
Santo Antão	Paul	11	2	9	8	2	6	123	231
	Porto Novo	23	5	26	14	4	11	112	201
	Ribeira Grande	18	2	18	8	1	7	125	251
São Vicente	São Vicente	14	8	16	13	6	15	105	354
São Nicolau	Tarrafal	27	0	30	25	0	28	125	302
	Ribeira Brava	56	2	68	14	1	19	103	240
Sal	Sal	6	0	7	1	0	1	129	282
Boavista	Boavista	13	0	18	4	0	5	107	246
Maio	Maio	25	24	9	22	21	9	101	324
Santiago	Tarrafal	21	6	21	16	0	21	120	274
	São Miguel	19	4	22	10	4	8	149	276
	Santa Catarina	8	3	5	4	1	3	117	282
	São Salvador do Mundo	21	4	21	17	4	16	121	319
	São Lourenço dos Órgãos	11	2	15	3	0	5	130	406
	Santa Cruz	3	1	3	2	1	1	142	359
	São Domingos	20	1	19	8	1	7	134	281
	Praia	11	2	12	7	1	7	140	386
	Ribeira Grande	4	0	4	2	0	3	108	208
Fogo	São Filipe	13	4	12	6	3	4	100	151
	Mosteiros	4	0	4	3	0	3	105	138
	Santa Catarina	1	0	3	0	0	0	100	242
Brava	Brava	35	2	44	29	2	35	116	360
**Total**		**364**	**72**	**386**	**218**	**55**	**209**	**2612**	**6113**

**Table 2 ijerph-17-01291-t002:** Positive containers for *Aedes aegypti*.

Container	No. of Containers with *Ae. aegypti* (%)	No. of *Ae. aegypti* (%)	No. of Containers with Other Species (%)
Ceramic pots	5 (1.9)	101 (2.5)	1 (0.5)
Buckets	10 (3.8)	102 (2.5)	2 (1.0)
Cisterns	8 (3.0)	107 (2.6)	6 (3.1)
Water drums	162 (61.4)	1904 (46.1)	53 (27.6)
Flowerpots	38 (14.4)	1455 (36.0)	9 (4.7)
Puddles	3 (1.1)	26 (0.6)	6(3.1)
Tanks	21 (8.0)	249 (6.2)	70 (36.5)
Tires	2 (0.8)	11 (0.3)	0 (0.0)
Other	8 (3.0)	47 (1.2)	45 (23.4)
Water fountains	7 (2.7)	43 (1.1)	0 (0.0)
**Total**	**264**	**4045**	**192**

**Table 3 ijerph-17-01291-t003:** Entomological indices in municipalities.

Island	Municipalities	Entomological Indices		
		HI (%)	CI (%)	BI
**Santo Antão**	Paul	6.5	3.5	6.5
	Porto Novo	12.5	7.5	13.4
	Ribeira Grande	6.4	6.0	12.0
**São Vicente**	São Vicente	12.4	5.9	20.0
**São Nicolau**	Tarrafal	20.0	9.27	22.4
	Ribeira Brava	13.6	8.3	19.4
**Sal**	Sal *	0.8	0.4	0.8
**Boavista**	Boavista *	3.7	2.0	4.7
**Maio**	Maio	21.8	9.3	29.7
**Santiago**	Tarrafal	13.3	7.7	17.5
	São Miguel	6.7	4.3	8.1
	Santa Catarina	3.4	1.4	3.4
	São Salvador do Mundo	14.1	6.3	16.5
	São Lourenço dos Órgãos *	2.3	1.2	3.8
	Santa Cruz *	1.4	0.6	1.4
	São Domingos	6.0	2.8	6.0
	Praia	5.0	2.1	5.7
	Ribeira Grande *	1.9	1.4	2.7
**Fogo**	São Filipe	6.0	4.6	7.0
	Mosteiros *	2.9	2.2	2.9
	Santa Catarina **	0.0	0.0	0.0
**Brava**	Brava	25.0	10.3	31.9

* Municipalities with Breteau Index (BI) <5; ** No *Ae. aegypti* larva found; ** *Ae. aegypti* adult mosquitoes were recorded during survey. House Index (HI), Container Index (CI).

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
