# Peer review of "Abundance and Updated Distribution of Aedes aegypti (Diptera: Culicidae) in Cabo Verde Archipelago: A Neglected Threat to Public Health"

_ijerph, 2020, doi:10.3390/ijerph17041291_

Round 1
Reviewer 1 Report
MINOR COMMENTS:
Line 18: Usually, the virus names have the following nomenclature, i.e. without capital latters: dengue, yellow fever, chikungunya. In contrast, Zika and West Nile are names of localities and thus written with capitals.
Line 21: see above
Line 31: change to: … most productive habitats found.
Line 40-41, 43: see above
Line 41: YF is not such a challenge as there is a vaccine available.
Line 45: First appearance of a species name should be accompanied by the authority, i.e.: Globally, Aedes aegypti (Linnaeus, 1762) [=Stegomyia aegypti] is…
Line 46: dito: …Aedes albopictus (Skuse, 1895) [=Stegomyia albopicta]…
Line 48: … caused most human…; …is a highly anthropophilic, peri-domestic….
Line 51: dengue
Line 51: 174-dengue???
Line 55: …associated with these…
Line 56: … is located at the west coast…
Lines 70-73: Aedes caspius (Pallas, 1771); Ae. aegypti; Anopheles…..; tigripes (de Grandpre and de Charmoy, 1901); and Culiseta longiareolata (Marcquart, 1838)…
Note: need brackets where the original description was under another generic name!
Line 95: change: In studies which did not observe this association, factors ….
Line 106: ..off the coast of Senegal (Figure 1). Note: delete the figure caption.
Line 107: … a total of about 537.600 inhabitants.
Line 114: In 2010, 141.762 accom….
Line 131: Swap: We selected the sampling area…
Lines 140, 142: reared
Paragraph lines 166ff.: This presentation does not correspond to the presentation in Fig. 2. Re-phrase!
Line 177: (Table 2) not 3!
Line 191: re-phrase: …the first archipelago-wide analysis on …. of Ae. aegypti for Cabo Verde, …
Lines 193, 198: see first comment
Line 204: …key factors…
Line 211: customs (not costumes)
Lines 216-17: …observed that Ae. aegypti extremely adapted to domestic habitats. (delete “uses….”)
Line 218: change: …indoor and outdoor…
Line 226: …, as well asresistance…
Author Response
Dear reviewer,
Thank your interest in our manuscript and the helpful corrections for its improvement. Your comments and suggestions have been taken into consideration point by point. Changes to the original document were registered in review mode in the re-submitted document.
Comments and Suggestions for Authors
MINOR COMMENTS:
Line 18: Usually, the virus names have the following nomenclature, i.e. without capital latters: dengue, yellow fever, chikungunya. In contrast, Zika and West Nile are names of localities and thus written with capitals.
Authors: Viruses names have been corrected accordingly throughout the document.
Line 21: see above
Authors: OK.
Line 31: change to: … most productive habitats found.
Authors: OK.
Line 40-41, 43: see above
Authors: OK.
Line 41: YF is not such a challenge as there is a vaccine available.
Authors: We agree, although other factors may contribute to a public health challenge on YF. However the emphasis should be on other arboviruses and we removed YF from the text.
Line 45: First appearance of a species name should be accompanied by the authority, i.e.: Globally, Aedes aegypti (Linnaeus, 1762) [=Stegomyia aegypti] is…
Authors: We agreed and completed.
Line 46: dito: …Aedes albopictus (Skuse, 1895) [=Stegomyia albopicta]…
Authors: We agreed and completed.
Line 48: … caused most human…; …is a highly anthropophilic, peri-domestic….
Authors: We agreed and changed text.
Line 51: dengue
Authors: OK.
Line 51: 174-dengue???
Authors: The hyphen was removed
Line 55: …associated with these…
Authors: OK.
Line 56: … is located at the west coast…
Authors: OK.
Lines 70-73: Aedes caspius (Pallas, 1771); Ae. aegypti; Anopheles…..; tigripes (de Grandpre and de Charmoy, 1901); and Culiseta longiareolata (Marcquart, 1838)…
Note: need brackets where the original description was under another generic name!
Authors: OK.
Line 95: change: In studies which did not observe this association, factors ….
Authors: OK.
Line 106: ..off the coast of Senegal (Figure 1). Note: delete the figure caption.
Authors: OK.
Line 107: … a total of about 537.600 inhabitants.
Authors: OK.
Line 114: In 2010, 141.762 accom….
Authors: OK.
Line 131: Swap: We selected the sampling area…
Authors: OK.
Lines 140, 142: reared
Authors: OK.
Paragraph lines 166ff.: This presentation does not correspond to the presentation in Fig. 2. Re-phrase!
Authors: The text was rephrased: “Water drums (50-200 L) were the most common breeding sites (62.9%; n=3,843), followed by tanks (1,000-5,000L) (12.4%; n=756), cisterns (5,000-10,000L) (6.3%; n=387) and other types of containers that comprised 18.4% of total breeding sites. From all inspected containers, 458 (7.5%) were positive, and the productivity proportion analysis for each container shown puddles as the most productive habitat (33%; n=9), followed by flowerpots (21%; n=47), others (18%; n=53), tanks (12%; n=91). All the other types of containers show productivity equal less than 10% (Figure 2)”
Line 177: (Table 2) not 3!
Authors: OK.
Line 191: re-phrase: …the first archipelago-wide analysis on …. of Ae. aegypti for Cabo Verde, …
Authors: Re-phrased
Lines 193, 198: see first comment
Authors: OK.
Line 204: …key factors…
Authors: OK.
Line 211: customs (not costumes)
Authors: OK.
Lines 216-17: …observed that Ae. aegypti extremely adapted to domestic habitats. (delete “uses….”)
Authors: OK.
Line 218: change: …indoor and outdoor…
Authors: OK.
Line 226: …, as well asresistance…
Authors: OK.
Please consider here the authors answers to your questions.
Looking forward to hearing from you.
On behalf of authors,
Silvânia Leal
Reviewer 2 Report
The work performed is quite impressive considering the logistics, amount of mosquitoes collectes and analysis.
Please, include a description for all tables, including key information that are in the main text.
Increase map picture quality, is quite pixelated and letter A should come before letter B - no need to have a lines delimiting this panel.
In figure 2, please change "negative" pattern for a grayscale color, the current hatching scheme is not appropriate. This figure also tend to say that puddles, flowerpots and other containers are the most frequent found positive, however the total numbers observed isn't quite ponderated here. adding a different type of graph divided by classes could improve the analysis and the way it is presented.
Please, all decimal numbers should be changed to dots instead of commas (main text and tables)
It would be interesting to see if there is any correlation among the islands in barlavento and sotavento, and their common characteristics, including the type of countainer.
Is there any correlation between the highiest mosquito density and dengue transmission?
Author Response
Dear reviewer,
Thank your interest in our manuscript and the helpful corrections for its improvement. Your comments and suggestions have been taken into consideration point by point. Changes to the original document were registered in review mode in the re-submitted document.
Comments and Suggestions for Authors
MINOR COMMENTS:
The work performed is quite impressive considering the logistics, amount of mosquitoes collectes and analysis.
Please, include a description for all tables, including key information that are in the main text.
Authors: we think the table caption is clear and it is unnecessary to repeat body text information. But we may not have understood this point well. Can you give an example?
Increase map picture quality, is quite pixelated and letter A should come before letter B - no need to have a lines delimiting this panel.
Authors: The map has been reworked (letters, lines and resolution).
In figure 2, please change "negative" pattern for a grayscale color, the current hatching scheme is not appropriate. This figure also tend to say that puddles, flowerpots and other containers are the most frequent found positive, however the total numbers observed isn't quite ponderated here. adding a different type of graph divided by classes could improve the analysis and the way it is presented.
Authors: Figure 2 has been changed (negative to grayscale colour and order – from more positive to more negative containers).
Please, all decimal numbers should be changed to dots instead of commas (main text and tables)
Authors: Ok.
It would be interesting to see if there is any correlation among the islands in barlavento and sotavento, and their common characteristics, including the type of countainer.
Authors: Our results indicate a significative correlation between barlavento/ sotavento and type of container (observed (X2=589.894, p<0.005), as also a positive association between the container type and the presence of Ae. aegypti (X2=133.816, p<0.001) (Line 180), but we did not further analyse other possible common characteristics between island’s groups.
Is there any correlation between the highiest mosquito density and dengue transmission?
Authors: In Cabo Verde (CV) a mosquito density threshold correlated to dengue transmission was not calculated yet. CV experience one dengue outbreak, and transmission is not currently taken place in Cabo Verde.
During the 2009 DEN-3 outbreak all islands, except Santo Antão and São Vicente, reported autochthone dengue cases, but mosquito density data at a regional level from that period is scarce. After this outbreak, the country has registered only sporadic cases of dengue, probably due to the decline in number of humans susceptible to this serotype. However, dengue serotypes 2 and 4 were already reported in Ae. aegypti populations (Guedes et al., 2017), which raises the concern for new outbreaks with these dengue serotypes. Anyway, only with long-term monitoring of Ae. aegypti populations at a regional level in CV will be possible to achieve the closest correlation between mosquito density and dengue epidemic transmission. Long-term monitoring of Ae. aegypti is one of our goals in the future for the public health of CV.
Guedes, D.R.; Gomes, E.T.; Paiva, M.H.; Melo-Santos, M.A.D.; Alves, J.; Gómez, L.F.; Ayres, C.F. Circulation of DENV2 and DENV4 in Aedes aegypti (Diptera: Culicidae) mosquitoes from Praia, Santiago Island, Cabo Verde. J. Insect Sci 2017, 17 (4), 86.
Please consider here the authors answers to your questions.
Looking forward to hearing from you.
On behalf of authors,
Silvânia Leal